# Prostaglandins in biofluids in pregnancy and labour: A systematic review

**Eilidh M. Wood[1], Kylie K. Hornaday[1], Donna M. Slater[1,2]***

**1** Department of Physiology and Pharmacology, Cumming School of Medicine, University of Calgary, Calgary, Alberta, Canada, **2** Department of Obstetrics and Gynecology, Cumming School of Medicine, University of Calgary, Calgary, Alberta, Canada

* dmslater@ucalgary.ca

## Abstract

Prostaglandins are thought to be important mediators in the initiation of human labour, however the evidence supporting this is not entirely clear. Determining how, and which, prostaglandins change during pregnancy and labour may provide insight into mechanisms governing labour initiation and the potential to predict timing of labour onset. The current study systematically searched the existing scientific literature to determine how biofluid levels of prostaglandins change throughout pregnancy before and during labour, and whether prostaglandins and/or their metabolites may be useful for prediction of labour. The databases EMBASE and MEDLINE were searched for English-language articles on prostaglandins measured in plasma, serum, amniotic fluid, or urine during pregnancy and/or spontaneous labour. Studies were assessed for quality and risk of bias and a qualitative summary of included studies was generated. Our review identified 83 studies published between 1968–2021 that met the inclusion criteria. As measured in amniotic fluid, levels of $PGE_2$, along with $PGF_{2\alpha}$ and its metabolite 13,14-dihydro-15-keto-$PGF_{2\alpha}$ were reported higher in labour compared to non-labour. In blood, only 13,14-dihydro-15-keto-$PGF_{2\alpha}$ was reported higher in labour. Additionally, $PGF_{2\alpha}$, $PGF_{1\alpha}$, and $PGE_2$ were reported to increase in amniotic fluid as pregnancy progressed, though this pattern was not consistent in plasma. Overall, the evidence supporting changes in prostaglandin levels in these biofluids remains unclear. An important limitation is the lack of data on the complexity of the prostaglandin pathway outside of the PGE and PGF families. Future studies using new methodologies capable of co-assessing multiple prostaglandins and metabolites, in large, well-defined populations, will help provide more insight as to the identification of exactly which prostaglandins and/or metabolites consistently change with labour. Revisiting and revising our understanding of the prostaglandins may provide better targets for clinical monitoring of pregnancies. This study was supported by the Canadian Institutes of Health Research.

## Introduction

It is widely believed that prostaglandins are important in the initiation of human labour [1]. Multiple studies have documented increased expression of cyclooxygenases, key enzymes in

**Data Availability Statement:** All relevant data are within the paper and its Supporting information files.

**Funding:** This study was supported by the Canadian Institutes of Health Research (CIHR) to

DMS (CIHR grant PJT-173295), a University of Calgary, Eye's High Doctoral Recruitment Scholarship to KKH and an NSERC Undergraduate Student Research Award to EMW. The funders had no role in study design, data collection and analysis, decision to publish, or preparation of the manuscript.

**Competing interests:** The authors have declared that no competing interests exist.

prostaglandin synthesis, in gestational tissues with the onset of labour, however, this has not been consistently observed [2]. Additionally, prostaglandins are present in maternal blood, urine, and amniotic fluid during pregnancy [3], however, the evidence supporting or refuting their role in labour is conflicting. Prostaglandins are known to affect uterine contractility and cervical ripening [4] and have thus been successfully used for labour induction since the late 1960's, though the use of prostaglandin synthesis inhibitors for prevention of preterm birth has been minimally successful and is associated with various fetal side effects [5]. Since their discovery in the 1930s, prostaglandins and their synthesis and metabolism are now known to be highly complex, which may contribute to these inconsistent outcomes seen during clinical targeting of this pathway. Aside from providing insight into labour processes, the presence of prostaglandins in peripheral tissues offers the potential for minimally invasive early prediction of labour onset and the ability to distinguish between true and false labour, which remains an ongoing clinical challenge [6]. Additionally, it has been suggested that biomarkers predictive of term labour (>37 weeks gestation) may also be useful for prediction of preterm labour (<37 weeks gestation), as both processes share common physiological changes involving cervical ripening, uterine contractions, and membrane rupture [7]. In 2010, preterm birth was estimated to occur in approximately 11% of all pregnancies and remains the leading cause of neonatal mortality worldwide [8], yet there is a lack of objective measures available to assess risk of premature delivery. Accurate prediction of term and preterm labour would allow for more informed patient planning and more efficient use of healthcare resources, for example, by reducing unnecessary hospitalizations and interventions. Despite evidence to suggest a role for prostaglandins in pregnancy and labour, literature defining the complexities of the pathway remain inconclusive and inconsistent. Therefore, we have systematically reviewed the scientific literature with the aim of answering three main questions to find evidence that either supports or refutes a role for prostaglandins in the initiation of labour: 1) Are prostaglandins or their metabolites detectable in biofluids in higher amounts in labour vs not in labour? 2) Are prostaglandins or their metabolites detected in increasing amounts prior to the onset of labour? And 3) Are prostaglandins or their metabolites present in urine, blood, or amniotic fluid predictive of preterm labour?

## Methods

This systematic review was conducted and reported following the recommendations of the Preferred Reporting Items for Systematic Review and Meta-Analyses (PRISMA). The protocol is available upon request. This review was not registered.

### Information sources

The databases MEDLINE and EMBASE were searched for records. Additionally, the reference lists of eligible studies and relevant review articles were manually searched.

### Search strategy

The search strategy included the key words "prostaglandins" AND "obstetric labor" AND ("amniotic fluid" OR "blood" OR "urine") as well as synonyms, related alternatives, and Medical Subject Heading (MeSH) terms as relevant. The searches were limited to human studies. Full details of the search terms for each database are given in S1 and S2 Tables. Citations retrieved from the initial search were downloaded into a reference manager (EndNote X9) and duplicates were removed. Two reviewers (EW and SLW) independently reviewed abstracts and removed those not relevant to the research questions. Following retrieval of full-text articles, both reviewers assessed the remaining citations against the eligibility criteria. Studies

excluded at this level were sorted based on reason for exclusion. Disagreements were resolved by discussion until consensus was reached.

## Inclusion/Exclusion criteria

Primary study journal articles examining endogenous prostaglandins in blood, amniotic fluid, and/or urine during pregnancy and spontaneous labour were included in this review. Studies were excluded if the study was on animals, the study was examining exogenous prostaglandins for induction of labour or if participants experienced spontaneous abortion (prior to 20 weeks). As well, studies which only had samples collected following delivery were excluded. Publications with incomplete information (i.e., conference abstracts) were excluded. Only studies written in English or with an available English translation were included. The search did not include a time restriction, however, the databases MEDLINE and EMBASE include literature published since 1946 and 1947 respectively. The search was initially conducted on May 19, 2020 and was repeated on August 20, 2021.

## Selection process/Data extraction

The following information was extracted by one reviewer (EW) from each of the final selected studies: population examined, sample number, type of biofluid collected, method of testing and measurement, metabolites/prostaglandins measured, time of sample collection, country of study origin, available measures of central tendency and variance, and major findings of the study.

## Quality assessment

Studies were assessed for quality and risk of bias using a quality assessment tool (Table 1) adapted from Hadley et al. [9] for assessment of basic science research. Full details of the rubric can be found in Table 1. Studies were scored between 0–9. All studies were scored independently by two investigators (EW and KH) and disagreements in scores were resolved by discussion.

**Table 1. Quality assessment rubric.**

| Quality assessment | 1 point | 0 points | N/A |
|---|---|---|---|
| 1) Question/objective sufficiently described? | Primary study question or objective is clearly stated | Unclear question/objective or no question/objective | |
| 2) Design appropriate to answer study question? | Study design is clearly stated and makes sense according to the study question/objective | E.g. uses convenient samples or study does not give enough information to determine study design | |
| 3) Methods described in sufficient detail to allow for study to be replicated? | Samples, reagents, assay used to measure prostaglandins are sufficiently described, methods for sample collection are clearly described | Some information missing or no information/ insufficient information is given on samples, reagents, assays, methods for sample collection | |
| 4) Researchers used blinding? | Yes | No | |
| 5) Sample number sufficient for internal validity? | Study has pre-planned sample size and/or power analysis or confidence intervals suggest sufficient sample size | No power analysis or confidence intervals suggest insufficient sample size | |
| 6) Appropriate negative controls? | Control group is appropriate to answer study question | Controls are from a clearly different population | |
| 7) Appropriate statistical analysis? | There is a comparison of means with appropriate transformations of data | No statistical analysis provided | |
| 8) Results reported in sufficient detail? | Results match methods i.e. all prostaglandins measured are reported on | Some measurements missing from results | |
| 9) Do the results support the conclusion? | Conclusion makes sense given results and answers primary study question/objective | Conclusion is overstated based on results or not related to main study question and main results | |

### Data synthesis

A qualitative summary was generated, and tables were created with the main results from each study. No pooled analysis was performed.

## Results

### Studies identified

The electronic search returned 2257 unique records after removal of duplicates from 2688 records. 2101 records were removed at the title/abstract level, leaving 156 records for assessment at the full-text level. Hand search of reference lists yielded an additional 35 records for review, resulting in a total of 191 full text records. Of the records assessed at the full text level, 108 were excluded, leaving n = 83 studies for inclusion in this review (Fig 1).

### Main characteristics of studies

Summaries of the main characteristics and relevant findings of the included studies can be found in Table 2 (presented in chronological order). Of the 83 studies, most assessed only one

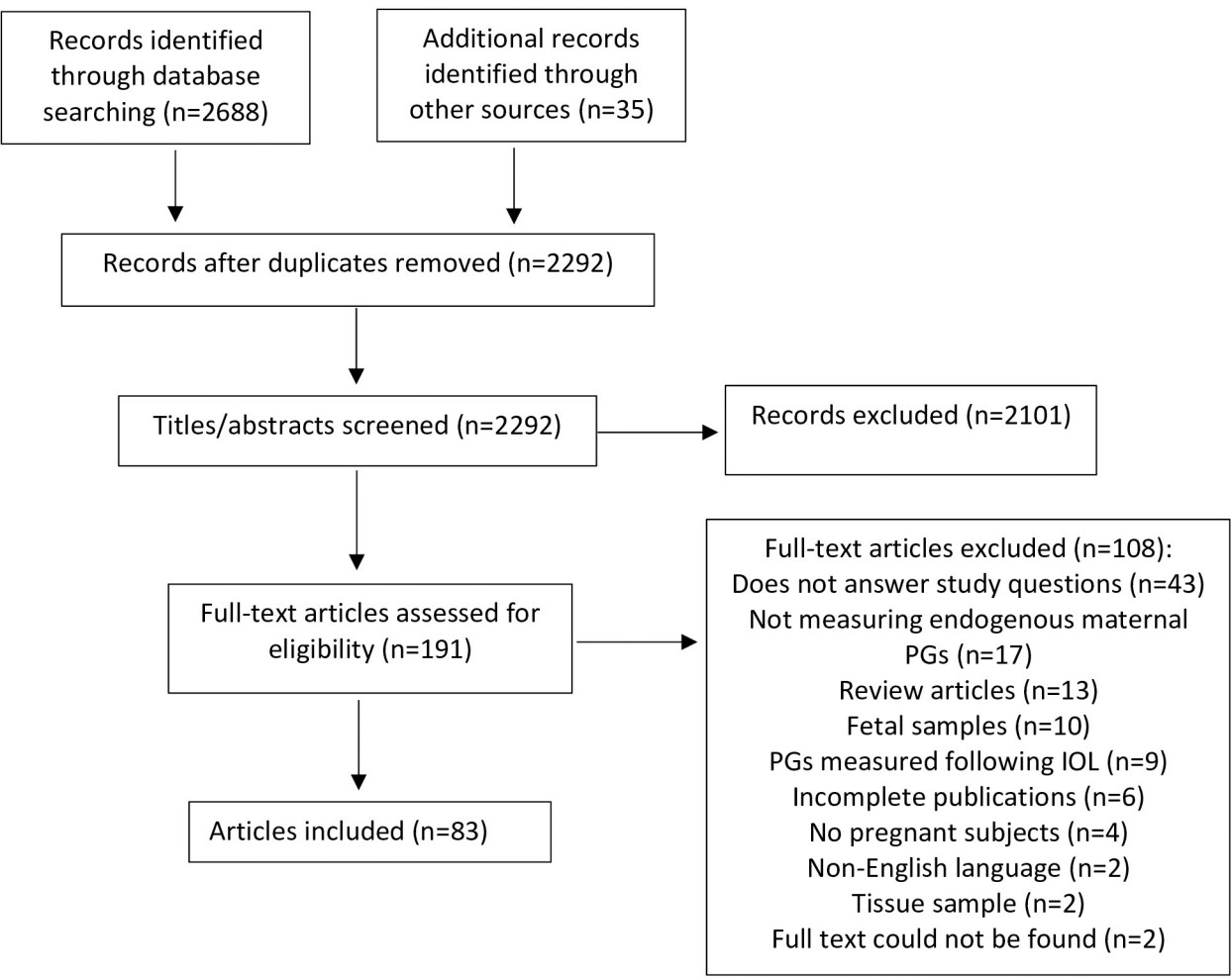

**Fig 1. PRISMA diagram.** Abbreviations: PG = prostaglandin, IOL = induction of labour.

**Table 2. Main characteristics of studies.**

| Study | Method | Sample Size | Biofluid | PG/Metabolite | Relevant Findings |
|---|---|---|---|---|---|
| Karim 1968 [10] | TLC and biological assay | n = 42 NL, n = 10 TL | plasma | $PGF_{2\alpha}$, $PGF_{1\alpha}$, $PGE_1$, $PGE_2$ | NL < LOD |
| | | | | $PGF_{2\alpha}$ | higher at delivery than 1st stage labour |
| Brummer 1972 [11] | RIA | n = 40 NL, n = 46 L | serum | $PGF_{2\alpha}$ | L>TNL |
| | | | | | late pregnancy similar to nonpregnant |
| | | | | | increased through 1st stage labour (early-late), then decreased in 2nd stage |
| Gutierrez-Cernosek & Levine 1972 [12] | RIA | n = 10 1st TM | serum | $PGF_{2\alpha}$ | peaked at 2nd TM, decreased to nonpregnant levels at term |
| | | n = 52 2nd TM | | | |
| | | n = 54 3rd TM | | | |
| | | n = 9 serial (14-40wks) | | | |
| Brummer 1973 [13] | unknown | n = 13 1st TM | serum | $PGF_{2\alpha}$ | decreased in 2nd TM, plateaued in 3rd TM |
| | | n = 40 2nd TM | | | |
| | | n = 75 3rd TM | | | |
| Brummer & Craft 1973 [14] | RIA | n = 58 L | serum | $PGF_{2\alpha}$ | highest in 1st stage labour, decreased in 2nd stage and remained low |
| | | n = 7 serial L | | | |
| Hertelendy et al 1973 [15] | RIA | n = 8 PTNL | plasma | PGE | <32wks pregnant similar to nonpregnant |
| | | n = 32 L | | | increased through 1st stage labour (early-late), then decreased in 2nd stage |
| Keirse & Turnbull 1973 [16] | GC | n = 12 TNL, n = 38 TL | AF | $PGE_2$ | TL>TNL |
| | | | | | increased through 1st stage labour |
| | | | | $PGE_1$ | <LOD |
| Salmon & Army 1973 [17] | RIA | n = 57 | AF | $PGF_{2\alpha}$ | L>TNL |
| | | | | | constant through 2nd and 3rd TM, rise after 36wks |
| | | | | | spike during 1st stage labour |
| Challis et al 1974 [18] | RIA | n = 4 TNL, n = 9 TL | plasma | PGF | TL>TNL (nonsignificant) |
| Green et al 1974 [19] | GC-MS | n = 2 TL | plasma | $PGF_{2\alpha}$ | no correlation with stage of labour |
| | | n = 5 term serial | | PGFM | TL>TNL |
| | | | | | increased through 1st stage labour |
| Hamberg 1974 [20] | RIA? | n = 3 serial (9-40wks) | urine | t-PGFM | increased with GA, peaked at term |
| | | n = 8 TNL, n = 1TL | | | TL>TNL |
| Hennam et al 1974 [21] | RIA | n = 13 1st TM | plasma | $PGF_{2\alpha}$ | levels lowest at 2nd TM compared to 1st and 3rd |
| | | n = 10 2nd TM | | | L>3rd TM |
| | | n = 20 3rd TM | | | |
| | | n = 99 L | | | |
| Hibbard et al 1974 [22] | RIA | n = 42 TNL, n = 13 TL | AF | $PGF_{2\alpha}$ | TL>TNL |
| | | | | | increased with GA after 36 weeks |
| | | n = 22 PTNL | | | 64% <LOD |
| Hillier et al 1974 [23] | RIA | n = 11 TNL, n = 5 TL | AF | $PGF_{2\alpha}$ | TL>TNL |
| | | | | | increased with labour stage |
| | | | plasma | | no correlation with labour stage |

(*Continued*)

**Table 2.** (Continued)

| Study | Method | Sample Size | Biofluid | PG/Metabolite | Relevant Findings |
|---|---|---|---|---|---|
| Keirse et al 1974 [24] | RIA and GC | n = 20 TNL, n = 26 TL | AF | PGF | TL>TNL |
| | | n = 8 PTNL | | | TNL>PTNL |
| | | | | | increased through 1st stage labour |
| MacDonald et al 1974 [25] | RIA | n = 6 NL, n = 6 L | AF | $PGF_{2\alpha}$ | L>NL |
| Singh & Zuspan 1974 [26] | PC and TD | n = 6 | AF | $PGF_{2\alpha}$, $PGF_{1\alpha}$, $PGE_1$, $PGE_2$ | constant from 24-36wks, increase in labour |
| Hillier et al 1975 [27] | RIA | n = 13 TNL, n =? TL | AF | PGF | increased with labour stage, peaked before delivery |
| | | n = 8 PTNL | | | increased from 2nd TM to term |
| Johnson et al 1975 [28] | RIA | n = 38 NL, n = 8 L | AF | $PGF_{2\alpha}$ | L>NL |
| | | n = 11 PTNL | | | 3rd TM > 2nd TM |
| | | n = 33 NL, n = 99 L | plasma | | no difference between NL and L |
| | | n = 15 PTNL | | | no pattern with labour |
| Pokoly & Jordan 1975 [29] | RIA | n = 6 TNL, n = 2 TL (CS) | AF | PGF | TL>TNL for CS only |
| | | n = 4 TNL, n = 13 TL | plasma | | no difference between NL and L |
| | | | AF | PGE | TL>TNL (nonsignificant) |
| | | | plasma | | no difference between NL and L |
| Dray & Frydman 1976 [30] | RIA | n = 24 NL, n = 37 L | AF | $PGF_{2\alpha}$ | L>NL |
| | | n = 19 PTNL | | | higher in late 3rd TM than early 3rd TM |
| | | | | | increased with labour stage |
| | | | | $PGE_2$ | L>TNL |
| | | | | | <LOD before 24wks, increased to 36wks, then remained constant to term |
| | | | | | increased with labour stage |
| | | | | $PGE_1$ | <LOD |
| Granstrom & Kindahl 1976 [31] | RIA | n = 1 term serial | urine | t-PGFM | TL>TNL |
| | | | | | late 3rd TM > nonpregnant |
| Keirse et al 1977 [32] | RIA | n = 40 TNL, n = 46 TL | AF | PGF, PGFM | TL>TNL |
| | | | | | increased through 1st stage labour |
| Kinoshita et al 1977 [33] | RIA | n = 7 TNL, n = 10 TL | AF | $PGF_{2\alpha}$ | TL>TNL |
| | | | | $PGE_1$ | no difference between TNL and TL |
| | | n = 10 TL, n = 10 TNL | plasma | $PGF_{2\alpha}$ | no difference between TNL and TL |
| | | n = 10 3rd TM serial | | | no pattern with gestation in 3rd TM |
| | | | | $PGE_1$ | no difference between TNL and TL |
| | | | | | no pattern with gestation in 3rd TM |
| | | | | | no correlation with labour stage |
| TambyRaja et al 1977 [34] | RIA | n = 27 PTL | AF | $PGF_{2\alpha}$ | increased through 1st stage labour |
| Haning et al 1978 [35] | RIA | n = 4 TNL, n = 8 TL | plasma | PGFM | TL>TNL |
| Mitchell et al 1978 [36] | RIA | n = 13 NL, n = 10 L | plasma | PGF, PGE | L > NL |
| | | n = 7 PTL | | | no correlation with stage of labour |
| | | | | PGFM | L > NL |
| | | | | | no difference with PTL and NL |
| | | | | | increased with labour stage |

(*Continued*)

**Table 2.** (Continued)

| Study | Method | Sample Size | Biofluid | PG/Metabolite | Relevant Findings |
|---|---|---|---|---|---|
| Nieder & Augustin 1978 [37] | RIA | n = 34 | AF | PGF$_{2\alpha}$, PGE | increased from 31wks to term, steeper after 36wks |
| | | | plasma | | no correlation with GA |
| Zuckerman et al 1978 [38] | RIA | n = 5 L | plasma | PGF$_{2\alpha}$ | lower in 1st stage labour than 2nd or 3rd |
| | | | | | peaked at delivery and at placental separation |
| Ghodaonkar et al 1979 [39] | RIA | n = 2 serial (20-40wks) | plasma | PGFM | no pattern with gestation |
| | | n = 14 TL | | | increased in 2nd and 3rd stages of labour |
| Mitchell et al 1979 [40] | RIA | n = 24 NL, n = 31 TL | AF | 6-keto-PGF$_{1a}$ | TL>TNL |
| | | | | | no correlation with GA or cervical dilation |
| Satoh et al 1979 [41] | RIA | n = 17 serial (8-39wks) | AF | PGFM | TL>TNL |
| | | | | | no pattern with gestation in 3rd TM |
| | | n = 8 TNL, n = 10 TL | plasma | | TL>TNL |
| | | n = 53 PTNL | | | no correlation with GA |
| | | | | | increased with labour stage, peaked at delivery |
| | | n = 30 3rd TM serial | urine | t-PGFM | L>NL |
| Lewis et al 1980 [42] | GC-MS | n = 6 1st TM | plasma | 6-keto-PGF$_{1a}$ | 2nd-3rd TM > nonpregnant |
| | | n = 9 2nd-3rd TM | | | |
| Dubin et al 1981 [43] | RIA | n = 39 serial (16-40wks) | plasma | PGFM | TL>TNL |
| | | n = 17 PTD | | | no correlation with GA |
| Sellers et al 1981 [44] | RIA | n = 13 TNL, n = 21 TL | plasma | PGFM | TL>TNL |
| | | n = 12 PTNL, n = 22 PTL | | | PTL>PTNL |
| | | | | | no difference between PTNL and PTL |
| | | | | | no difference between PTL who delivered term and preterm |
| | | | | | increased with labour stage in PTL and TL |
| Ylikorkala et al 1981 [45] | RIA | n = 9 serial | plasma | 6-keto-PGF$_{1a}$ | TL>TNL |
| | | | | | increased with labour stage |
| Fuchs et al 1982 [46] | RIA | n = 14 TNL, n = 20 TL | plasma | PGFM | TL>TNL |
| Fuchs et al 1982 [47] | RIA | n = 10 TNL, n = 14 TL | plasma | PGFM | TNL>PTNL |
| | | n = 10 PTNL, n = 15 PTL | | | PTL>PTNL |
| Mitchell et al 1982 [48] | RIA | n = 10 TNL, n = 10 TL | plasma | bicyclo-PGEM | TL>TNL |
| | | n = 10 1st TM | | | 1st TM > nonpregnant |
| | | n = 10 2nd TM | | | decreased in 3rd TM until labour |
| | | n = 10 3rd TM | | | |
| Sellers et al 1982 [49] | RIA | n = 10 TL | plasma | PGFM | increased with labour stage, peaked 5min after delivery |
| Sharma et al 1982 [50] | RIA | n = 92 NL, n = 6 TL | plasma | PGF$_{2\alpha}$ | TL>NL |
| | | | | | remained unchanged until 2wks before delivery, then increased |
| | | | | PGE$_2$ | no difference between TL and NL |
| | | | | | remained unchanged until 2wks before delivery, then increased |

(*Continued*)

**Table 2.** (Continued)

| Study | Method | Sample Size | Biofluid | PG/Metabolite | Relevant Findings |
|---|---|---|---|---|---|
| Fuchs et al 1983 [51] | RIA? | n = 4 TNL, n = 17 L | plasma | PGFM | TL>TNL |
| | | | | | increased with labour stage |
| Nieder & Augustin 1983 [52] | RIA | n = 23 1st TM | AF | PGF, PGE | unchanged from 9-34wks, increase at 35wks |
| | | n = 37 2nd TM | | | |
| | | n = 103 3rd TM | | | |
| Spitz et al 1983 [53] | RIA | n = 12 serial (10-40wks) | plasma | 6-keto-PGF$_{1a}$ | decrease after 33wks |
| Husslein & Sinzinger 1984 [54] | RIA | n = 5 TNL, n = 5 TL | plasma | PGEM | TL>TNL |
| | | n = 5 PTNL | | | no correlation with labour stage |
| Nagata et al 1984 [55] | RIA | n = 6 term serial | plasma | PGF$_{2\alpha}$ | TL>TNL |
| | | | | PGE$_1$, PGE$_2$ | no difference between NL and L |
| | | | | | no correlation with labour stage |
| Reddi et al 1984 [56] | RIA | n = 10 TL | AF | PGF, PGFM | increased through 1st stage labour |
| Sellers et al 1984 [57] | RIA | n = 14 TNL, n = 9 TL | plasma | PGFM | TL>TNL |
| Yamaguchi & Mori 1984 [58] | RIA | n = 4 <20wks | plasma | PGFM | L>NL |
| | | n = 3 20-30wks | | | no correlation with GA |
| | | n = 16 30-40wks | | 6-keto-PGF$_{1a}$ | L>NL (nonsignificant) |
| Brennecke et al 1985 [59] | RIA | n = 9 TNL, n = 27 TL | plasma | PGFM | TL>TNL |
| | | n = 12 serial | | | increased with labour stage |
| | | | | bicyclo-PGEM | no difference between TNL and TL |
| | | | | | no correlation with GA or labour stage |
| Ogino & Jimbo 1986 [60] | RIA | n = 5 24-28wks | plasma | PGF$_{2\alpha}$ | peak at 32-36wks |
| | | n = 4 28-32wks | | PGE$_2$ | lowest at 36-40wks |
| | | n = 7 32-36wks | | | |
| | | n = 8 36-40wks | | | |
| Weitz et al 1986 [61] | RIA | n = 6 PTL-TD | plasma | PGFM | PTL>PTNL |
| | | n = 14 PTL-PTD | | | higher in PTL who delivered PT than those who delivered term |
| | | n = 11 PTNL | | | |
| Ylikorkala et al 1986 [62] | RIA | n = 8 TNL, n = 13 TL | urine | 6-keto-PGF$_{1a}$ | increased with labour stage and with C-section |
| Berryman et al 1987 [63] | RIA | n = 23 L | AF | PGD$_2$ | increased through 1st stage labour |
| Nagata et al 1987 [64] | RIA | n = 9 TL | plasma | PGFM | increased with labour stage (nonsignificant) |
| | | | | PGE$_1$ | low throughout labour |
| Nagata et al 1987 [65] | RIA | n = 7 serial | plasma | PGFM | TL>TNL |
| | | | | | decreased 2wks prior to labour |
| | | | | | increased with labour stage |
| Romero et al 1987 [66] | RIA | n = 23 PTNL, n = 30 PTL | AF | PGF$_{2\alpha}$, PGE$_2$ | PTL>PTNL |
| Noort et al 1988 [67] | RIA | n = 12 1st TM | urine | 6-keto-PGF$_{1a}$ | L>NL (nonsignificant) |
| | | n = 12 2nd TM | | | |
| | | n = 12 3rd TM | | | |
| | | n = 12 TL | | | |
| Romero et al 1988 [68] | RIA | n = 32 PTL-TD n = 22 PTL-PTD | AF | PGE$_2$ | higher in PTL who did not respond to tocolysis than those who responded to tocolysis |

*(Continued)*

**Table 2.** (Continued)

| Study | Method | Sample Size | Biofluid | PG/Metabolite | Relevant Findings |
|---|---|---|---|---|---|
| Sahmay et al 1988 [69] | RIA | n = 8 TNL, n = 9 TL | AF | $PGF_{2\alpha}$ | no difference between TNL and TL |
| | | | plasma | | TL>TNL |
| | | | AF | PGE | no difference between TNL and TL |
| | | | plasma | | TNL>TL |
| Noort et al 1989 [70] | RIA | n = 7 TL | plasma | PGFM | increased with labour stage |
| | | | | 6-keto-PGF1a | no correlation with labour stage |
| Romero et al 1989 [71] | RIA | n = 25 PTL-TD | AF | $PGF_{2\alpha}$ | no difference between PTL who delivered term and preterm |
| | | n = 16 PTL-PTD | | PGFM, bicyclo-PGEM | higher in PTL who delivered PT than those who responded to tocolysis |
| Yamamoto & Kitao 1989 [72] | RIA | n = 76 term serial | plasma | $PGF_{2\alpha}$ | TL>TNL |
| | | | | | increased with labour stage and delivery |
| Mazor et al 1990 [73] | RIA | n = 10 PTL-TD | AF | $PGF_{2\alpha}$ | no difference between PTL who delivered term and preterm |
| | | n = 10 PTL-PTD | | $PGE_2$ | higher in PTL who delivered PT than those who delivered at term |
| Norman & Reddi 1990 [74] | RIA | n = 54 TL | AF | $PGF_{2\alpha}$, PGFM, $PGE_2$ | increased through 1st stage labour |
| Fairlie et al 1993 [75] | RIA | n = 20 TL | plasma | PGFM | increased with labour stage |
| | | | | bicyclo-PGEM | in nulliparous: rose after amniotomy but did not change with labour |
| | | | | | in multiparous: rose with amniotomy then increased with labour stage |
| Hillier et al 1993 [76] | RIA | n = 50 PTL | AF | $PGE_2$ | high levels associated with PTD and delivery within 1wk of amniocentesis |
| Johnston et al 1993 [77] | RIA | n = 18 TNL, n = 28 TL | plasma | PGFM | TL>TNL |
| | | | | | rose following amniotomy, then remained constant until delivery |
| | | | | PGEM | TL>TNL only in primigravid |
| | | | | | rose 1–2 after amniotomy, then remained constant until delivery |
| MacDonald & Casey 1993 [78] | RIA | n = 50 TNL, n = 190 TL | AF | $PGF_{2\alpha}$ | TL>TNL (forebag and upper compartment) |
| | | | | | increased with labour stage, then decreased at 3-5cm dilation |
| | | | | PGFM | TL>TNL (forebag and upper compartment) |
| | | | | | increased with labour stage, then leveled out at 4–5.5cm dilation until delivery |
| | | | | $PGE_2$ | TL>TNL (forebag) |
| | | | | | no difference between TL and TNL in upper compartment |
| | | | | | increased with labour stage, then leveled out at 4–5.5cm dilation until delivery |
| Romero et al 1993 [79] | RIA | n = 24 NL, n = 16 TL | AF | $PGF_{2\alpha}$, PGFM, $PGE_2$, 6-keto-$PGF_{1\alpha}$ | TL>NL |
| Romero et al 1994 [80] | RIA | n = 82 TNL, n = 168 TL | AF | $PGF_{2\alpha}$, PGFM, $PGE_2$, 6-keto-$PGF_{1\alpha}$ | TL>TNL |
| | | | | | increased through 1st stage labour |
| Lindsay et al 1995 [81] | ELISA | n = 8 serial (1st-3rd TM) | urine | 2,3-dinor-6-keto-$PGF_{1\alpha}$ | no correlation with GA |
| | | | | | pregnant >> nonpregnant |

(*Continued*)

**Table 2.** (Continued)

| Study | Method | Sample Size | Biofluid | PG/Metabolite | Relevant Findings |
|---|---|---|---|---|---|
| Romero et al 1996 [82] | RIA | n = 28 serial (n = 17 L) | AF | $PGF_{2\alpha}$, $PGE_2$ | TL>TNL |
| | | | | | increased with GA at term |
| Ichikawa & Minami 1999 [83] | RIA | n = 30 serial | urine | $PGF_{2\alpha}$ | TL>NL |
| | | | | | increased from 28-36wks |
| | | | | PGFM | TL>NL |
| | | | | | increased from 28-36wks and again at 2nd stage of labour |
| Mitchell et al 2005 [84] | ELISA | n = 24 TNL, n = 37 TL | AF | $9\alpha,11\beta$-$PGF_2$ | TL>TNL |
| | | n = 13 PTNL, n = 56 PTL | | | PTNL>PTL |
| Lee et al 2008 [85] | ELISA | n = 68 TNL, n = 34 TL | AF | $PGF_{2\alpha}$ | TL>TNL |
| | | n = 65 PTNL | | | no correlation with GA until 36wks, 25-fold increase at TNL |
| | | | | | increased with labour stage |
| | | | | $PGE_2$ | no difference between TL and TNL |
| | | | | | no correlation with GA until 36wks, 2-fold increase at TNL |
| Lee et al 2009 [86] | ELISA | n = 140 PPROM (n = 126 PTD) | AF | $PGF_{2\alpha}$ | high levels associated with low GA at delivery and PTD |
| Maddipati et al 2014 [87] | LC-MS | n = 10 TNL, n = 35 TL | AF | $PGF_{2\alpha}$, PGFM, $PGE_2$, bicyclo-PGEM, $PGA_2$, $PGJ_2$ | TL>TNL |
| | | n = 18 PTNL | | | |
| | | | | 19-OH-$PGE_2$ | no difference between TL and TNL |
| | | | | | TNL>PTNL |
| Park et al 2016 [88] | ELISA | n = 132 PTL (n = 41 PTD) | AF | $PGF_{2\alpha}$ | high levels associated with low GA at delivery and PTD |
| Rosen et al 2019 [89] | GC-NICI-MS | n = 740 (n = 41 sPTD) | urine | $PGF_{2\alpha}$ | no difference in 3rd TM levels between term and preterm delivery |
| Eick et al 2020 [90] | GC-NICI-MS | n = 469 (n = 50 PTD) | urine | $PGF_{2\alpha}$ | levels at 20-24wks and 24-28wks higher in preterm than term group |
| | | | | | associated with increased odds of PTB |
| Peiris et al 2020 [91] | LC-MS | n = 10 TNL, n = 28 TL | AF | $PGF_{2\alpha}$, PGFM, $PGE_2$ | TL>TNL |
| Takahashi et al 2021 [92] | LC-MS | n = 11 TNL, n = 10 TL | AF | $PGE_2$, 15-keto-$PGE_2$, PGEM, 19-OH-$PGE_2$ | TL>TNL |

Abbreviations: TLC = thin layer chromatography, NL = no labour, TL = term labour, LOD = limit of detection, RIA = radioimmunoassay, L = labour, TM = trimester, PTNL = preterm no labour, GC = gas chromatography, AF = amniotic fluid, TNL = term no labour, GC-MS = gas chromatography-mass spectrometry, PGFM = 13,14-dihydro-15-keto-$PGF_{2\alpha}$, t-PGFM = $5\alpha,7\alpha$-dihydroxy 11-keto tetranor-prostane 1,16-dioic acid, GA = gestational age, PC = paper chromatography, TD = transmission densitometry, CS = Caesarean section, PTL = preterm labour, PTD = preterm delivery, bicyclo-PGEM = 11-deoxy-13,14-dihydro-15-keto-11,16-bicyclo $PGE_2$, PGEM = 13,14-dihydro-15-keto-$PGE_2$, PTL-TD = preterm labour-term delivery, PTL-PTD = preterm labour-preterm delivery, PT = preterm, ELISA = enzyme-linked immunosorbent assay, PPROM = preterm premature rupture of membranes, LC-MS = liquid chromatography-mass spectrometry, NICI = negative ion chemical ionization, sPTD = spontaneous preterm delivery.

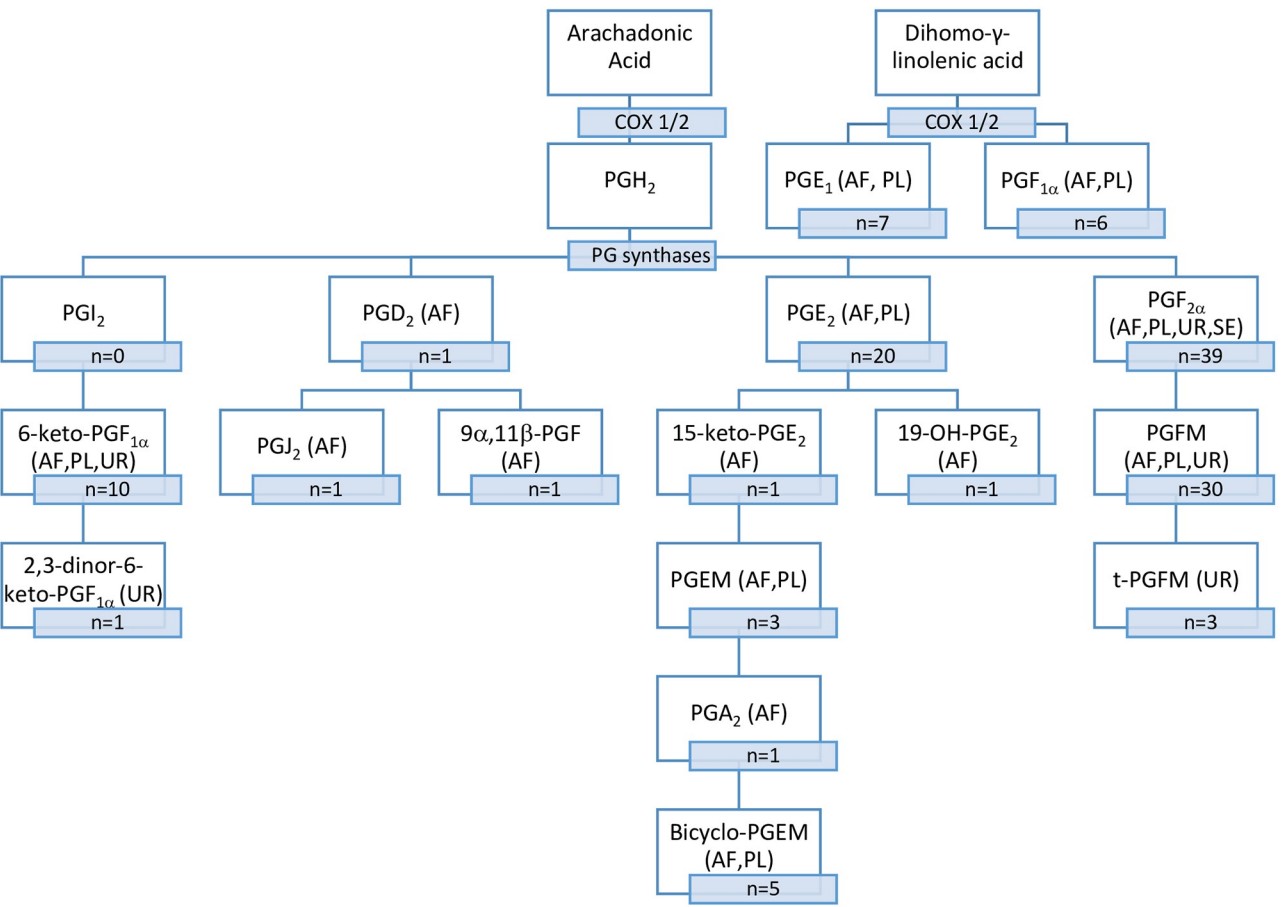

**Fig 2. Prostaglandin metabolism pathway.** n denotes the number of studies that measured the prostaglandin/metabolite. Abbreviations: AF = amniotic fluid, PL = plasma, UR = urine, SE = serum, COX 1/2 = cyclooxygenase 1/2, PGFM = 13,14-dihydro-15-keto-PGF$_{2\alpha}$, PGEM = 13,14-dihydro-15-keto-PGE$_2$, bicyclo-PGEM = 11-deoxy-13,14-dihydro-15-keto-11,16-bicyclo PGE$_2$, t-PGFM = 5$\alpha$,7$\alpha$-dihydroxy-11-keto tetranor-prostane-1,16-dioic acid.

biofluid, (34 plasma, 32 amniotic fluid, 8 urine, 4 serum) while 6 studies assessed multiple bio-fluids. The range of prostaglandins and metabolites investigated included PGF$_{2\alpha}$, PGF$_{1\alpha}$, 13,14-dihydro-15-keto-PGF$_{2\alpha}$ (PGFM), 5$\alpha$,7$\alpha$-dihydroxy-11-keto-tetranor-prostane-1,16-dioic acid (t-PGFM), PGE$_1$, PGE$_2$, 15-keto-PGE$_2$, 13,14-dihydro-15-keto-PGE$_2$ (PGEM), 11-deoxy-13,14-dihydro-15-keto-11,16-bicyclo-PGE$_2$ (bicyclo-PGEM), 6-keto-PGF$_{1\alpha}$, 2,3-dinor-6-keto-PGF$_{1\alpha}$, PGA$_2$, PGD$_2$, PGJ$_2$, 19-OH-PGE$_2$, and 9$\alpha$,11$\beta$-PGF$_2$. Prostaglandins and the corresponding metabolites measured are described in Fig 2. In addition, many older studies used measurement techniques which were unable to differentiate between subcategories of prostaglandins and therefore reported levels of PGE or PGF. The range of prostaglandin concentrations reported using different measurement techniques are shown in Table 3.

## Amniotic fluid

**Labour and non-labour.** In total, 25 studies compared amniotic fluid prostaglandins in labour vs non-labour. PGF$_{2\alpha}$ increased in labouring participants compared to non-labouring participants in most studies, (Table 2) however, one study found no difference [69]. Similarly, PGE$_2$ was reported to increase with labour in 11/12 studies [16, 26, 30, 66, 78–80, 82, 87, 91,

**Table 3. Range of prostaglandin concentrations reported using different measurement techniques.**

| Biofluid | PG/Metabolite | Measurement Technique | Range |
|---|---|---|---|
| plasma | PGE | RIA | NL: 4.8 [36]–3641.2 [69] (pg/ml), L: 5.4 [36]–2429.1 [69] (pg/ml) |
| | PGF | RIA | NL: 6.2 [36]–480 [29] (pg/ml), L: 7.9 [36]–600 [29] (pg/ml) |
| | PGE$_2$ | TLC and biological assay | NL: <200 pg/ml [10] |
| | | RIA | NL: 4.6 [60]–15,600 [55] (pg/ml), L: 559 [50]–21,200 [55] (pg/ml) |
| | PGE$_1$ | TLC and biological assay | <200 pg/ml [10] |
| | | RIA | NL: 2600 [55]–10,000 [33] (pg/ml), L: 4500 [64]–6800 [33] (pg/ml) |
| | PGF$_{2\alpha}$ | TLC and biological assay | NL: <200 pg/ml [10], L: <200 [10]– 18,000 [10] (pg/ml) |
| | | GC-MS | NL: <70 [19]–600 [19] (pg/ml), L: <100 [19]–200 [19] (pg/ml) |
| | | RIA | NL: 17 [37]–4600 [55] (pg/ml), L: 33.1 [21]–7900 [55] (pg/ml) |
| | PGF$_{1\alpha}$ | TLC and biological assay | NL: <200 pg/ml [10] |
| | 6-keto-PGF$_{1\alpha}$ | GC-MS | NL: 131 [42]–244 [42] (pg/ml) |
| | | RIA | NL: 18.7 [53]–318.6 [58] (pg/ml), L: 21 [70]–608 [70] (pg/ml) |
| | PGEM | RIA | NL: 58 [54]–554 [77] (pg/ml), L: 82 [54]–433 [77] (pg/ml) |
| | Bicyclo-PGEM | RIA | NL: 49 [48]–200 [59] (pg/ml), L: 62 [48]–500 [75] (pg/ml) |
| | PGFM | GC-MS | NL: 31 pg/ml [19], L: 267–942 [19] (pg/ml) |
| | | RIA | NL: 32.1 [61]–490 [41] (pg/ml), L: 20 [75]–2880 [58] (pg/ml) |
| serum | PGF$_{2\alpha}$ | RIA | NL: 200 [13]–1800 [12] (pg/ml), L: 100 [11]–3000 [11] (pg/ml) |
| AF | PGE | RIA | NL: 89 [37]–1400 [29] (pg/ml), L: 502.8 [69]–8800 [29] (pg/ml) |
| | PGF | RIA | NL: 50 [27]–1650 [24] (pg/ml), L: 500 [27]–75,000 [27] (pg/ml) |
| | PGE$_2$ | GC | NL: <200 [16]–6200 [16] (pg/ml), L: 1200 [16]–17,000 [16] (pg/ml) |
| | | PC and TD | NL: 250 [26]–300 [26] (pg/ml), L: 1700 pg/ml [26] |
| | | RIA | NL: <10 [30]–11,177 [80] (pg/ml), L: 17.8 [76]–28,197 [74] (pg/ml) |
| | | ELISA | NL: 24 [85]–4749 [85] (pg/ml), L: 62 [85]–36,651 [85] (pg/ml) |
| | | LC-MS | NL: <10 [87]–70,493 [87] (pg/ml), L: <10 [87]–105,739 [87] (pg/ml) |
| | PGE$_1$ | GC | NL: <500 pg/ml [16], L: <500 pg/ml [16] |
| | | PC and TD | NL: 1000 [26]–1200 [26] (pg/ml), L: 1800 pg/ml [26] |
| | | RIA | NL: <10 [30]–5000 [33] (pg/ml), L: 4400 pg/ml [33] |
| | PGF$_{2\alpha}$ | RIA | NL: 29 [66]–4700 [26] (pg/ml), L: 27 [71]–44,270 [33] (pg/ml) |
| | | ELISA | NL: 8 [85]–926 [85] (pg/ml), L: 78 [85]–15,326 [85] (pg/ml) |
| | | LC-MS | NL: <10 [87]–127 [91] (pg/ml), L: <10 [87]–42,537 [87] (pg/ml) |
| | PGF$_{1\alpha}$ | PC and TD | NL: 1500 [26]–2000 [26] (pg/ml), L: 12,000 pg/ml [26] |
| | 6-keto-PGF$_{1\alpha}$ | RIA | NL: 67 [79]–809 [80] (pg/ml), L: 68 [80]–1927 [80] (pg/ml) |
| | PGEM | LC-MS | NL: 71 pg/ml [92], L: 8425 pg/ml [92] |
| | Bicyclo-PGEM | RIA | L: 75 [71]- 4275 [71] (pg/ml) |
| | | LC-MS | NL: <10 [87]–66,900 [87] (pg/ml), L: 8361 [87]–133,800 [87] (pg/ml) |
| | 15-keto-PGE$_2$ | LC-MS | NL: 0 pg/ml [92], L: 210.24 pg/ml [92] |
| | 19-OH-PGE$_2$ | LC-MS | NL: 0 [92]–221,100 [87] (pg/ml), L: 73.7 [92]–202,675 [87] (pg/ml) |
| | PGFM | RIA | NL: 80 [79]–1571 [79] (pg/ml), L: 105 [71]–25,028 [56] (pg/ml) |
| | | LC-MS | NL: <10 [87]–114.79 [91] (pg/ml), L: <10 [87]–28,360 [87] (pg/ml) |
| | PGD$_2$ | RIA | L: 900 [63]–1800 [63] (pg/ml) |
| | PGJ$_2$ | LC-MS | L: 8542.8 pg/ml [87] |
| | 9α,11β-PGF$_2$ | ELISA | NL: 30 [84]–204 [84] (pg/ml), L: 10 [84]–396 [84] (pg/ml) |
| | PGA$_2$ | LC-MS | NL: <10 [87]–16,722 [87] (pg/ml), L: <10 [87]–50,167 [87] (pg/ml) |

(*Continued*)

**Table 3.** (Continued)

| Biofluid | PG/Metabolite | Measurement Technique | Range |
|---|---|---|---|
| urine | PGF$_{2\alpha}$ | RIA | NL: 0.99 [83]–1.85 [83] (pg/g creatinine), L: 2.03 [83]–3.14 [83] (pg/g creatinine) |
| | | GC-NICI-MS | NL: 1840 [90]–2060 [89] (pg/ml) |
| | 6-keto-PGF$_{1\alpha}$ | RIA | NL: 114,000 [67]–571,000 [67] (pg/g creatinine), L: 426,980 [62]–1,219,000 [67] (pg/g creatinine) |
| | PGFM | RIA | NL: 1.82 [83]–4.87 [83] (pg/g creatinine), L: 7.93 [83]–12.70 [83] (pg/g creatinine) |
| | t-PGFM | RIA | NL: 0.46 [20]–2.32 [20] (μg/hr), L: 1.06 [41]–2.50 [31] (μg/hr) |
| | 2,3-dinor-6-keto-PGF$_{1\alpha}$ | ELISA | NL: 623,232 [81]–1,096,181 [81] (pg/ml) |

Published data presented as ng/ml, nanomolars, or picomolars were converted to pg/ml and data presented as ng/g creatinine or ng/mmol creatinine were converted to pg/g creatinine. Data published in μg/hr were not converted and are presented as in the original article.

Abbreviations: RIA = radioimmunoassay, NL = non-labour, L = labour, TLC = thin layer chromatography, GC-MS = gas chromatography-mass spectrometry, PGEM = 13,14-dihydro-15-keto-PGE$_2$, bicyclo-PGEM = 11-deoxy-13,14-dihydro-15-keto-11,16-bicyclo PGE$_2$, PGFM = 13,14-dihydro-15-keto-PGF$_{2\alpha}$, PC = paper chromatography, TD = transmission densitometry, ELISA = enzyme-linked immunosorbent assay, LC-MS = liquid chromatography-mass spectrometry, GC-NICI-MS = gas chromatography-negative ion chemical ionization-mass spectrometry, t-PGFM = 5α,7α-dihydroxy 11-keto tetranor-prostane 1,16-dioic acid.

92]. 6-keto-PGF$_{1\alpha}$ and PGFM and were reported to increase in labouring participants compared to non-labouring participants in three [40, 79, 80] and seven [32, 41, 78–80, 87, 91] studies, respectively. Results were mixed for PGE$_1$ [26, 33]. PGE was not found to increase with labour [29, 69].

**Prior to labour onset.** Of the included amniotic fluid studies, 15 measured prostaglandins at more than one time point throughout pregnancy. PGF$_{2\alpha}$, PGF$_{1\alpha}$, and PGF were generally found to increase around term or prior to labour [17, 22, 27, 28, 30, 37, 52, 82, 85], though two studies found no pattern throughout pregnancy [26, 33]. Among studies that measured PGE or PGE$_2$, most (4/6) reported increased levels around 35–36 weeks [30, 37, 52, 85]. PGE$_1$ was not found to change with gestational age [26, 30, 33].

**Predicting preterm labour.** Six amniotic fluid studies investigated prostaglandins as predictors of preterm labour. Some studies suggest that PGF$_{2\alpha}$ may be predictive of preterm delivery in those with threatened preterm labour [88] and PPROM [86] however, results are mixed [71, 73]. PGFM and bicyclo-PGEM were found in higher levels in participants with preterm labour leading to preterm delivery compared to those who eventually delivered at term [71]. Increased PGE$_2$ levels may be predictive of delivery before term [68, 73] and before 34 weeks [76].

## Blood

**Labour and non-labour.** In total, 27 studies compared labour and non-labour groups. PGF$_{2\alpha}$ was reported to increase with labour compared to non-labour in most (6/8) studies [21, 41, 50, 55, 69, 72]. All 15 studies that measured PGFM reported higher levels in labour compared to non-labour (Table 2). Three studies measuring PGE reported varying results [29, 36, 69]. PGE$_1$, PGE$_2$, and PGF were all generally found to remain unchanged with labour [29, 33, 36, 50, 55].

**Prior to labour onset.** In total, 18 studies obtained measurements of plasma throughout pregnancy. Among those that measured PGF$_{2\alpha}$, some found increasing levels at or near term [37, 50, 60] however results were conflicting [21, 28, 33]. In 5/6 studies PGFM was not found to change with increasing gestational age [39, 41, 43, 44, 58]. Results for PGF$_{2\alpha}$ in serum were mixed [11–13].

**Predicting preterm labour.** Two studies investigated prostaglandins in plasma as predictors of preterm labour. One study found that PGFM levels were higher in participants in

preterm labour who delivered preterm compared to those who went on to deliver at term [61] though the other study reported no significant difference [44].

### Urine

**Labour and non-labour.**  Five studies compared labouring vs non-labouring groups. The metabolite t-PGFM was reported to increase with labour compared to non-labour [20, 31, 41].

**Prior to labour onset.**  Four studies measured changes in prostaglandins in urine throughout pregnancy. $PGF_{2\alpha}$, PGFM, and t-PGFM were reported to increase around term, though this was only reported in one study for each prostaglandin/metabolite [20, 83]. The metabolite 2,3-dinor-6-keto-$PGF_{1\alpha}$ did not change throughout pregnancy [81].

**Predicting preterm labour.**  Two urine studies investigated prostaglandins as predictors of preterm labour. One found that $PGF_{2\alpha}$ levels in urine samples collected at median 32.1 weeks were not significantly different between participants who delivered at term and those that delivered preterm [89]. In contrast, averaged levels of $PGF_{2\alpha}$ in urine were also found to be associated with increased odds of preterm birth (OR = 1.98) [90].

### Serial prostaglandin measurement during spontaneous labour

Although not a primary study question of this review, we noted that n = 40 studies obtained serial samples during labour. In general, prostaglandins measured in amniotic fluid increased throughout labour. Results were mixed for those that measured plasma and serum.

### Quality assessment

Scores from the quality assessment were distributed as follows: 17% scored between 0–3, 41% scored between 4–6, and 42% scored between 7–9. The areas with the lowest scores were researcher blinding and sufficiency of sample number for internal validity. Scores for each study can be found in S3 Table.

## Discussion

We demonstrate, through a systematic review of the literature investigating prostaglandins and metabolites in peripheral biofluids during pregnancy and labour, that prostaglandins of the PGE and PGF families do exhibit changes through pregnancy and labour, though results are inconsistent and inconclusive. Changes in $PGE_2$, $PGF_{2\alpha}$, and PGFM levels with labour are most prominent in amniotic fluid, and to a lesser extent in blood. Similarly, our synthesis suggests that $PGE_2$, $PGF_{2\alpha}$ and $PGF_{1\alpha}$ increase in amniotic fluid as pregnancy progresses and peak around term, though in plasma, a consistent pattern is unclear. Patterns in urine prostaglandin levels were inconclusive due to a relatively small number of studies investigating this biofluid. An important limitation is a general lack of data on prostaglandins and metabolites outside the PGE and PGF families, and as such we are unable to comment on their potential role in pregnancy and labour. Further, few studies examined prostaglandins as biomarkers for preterm labour and more research is needed to provide conclusive evidence for which prostaglandins or metabolites examined could offer the best options for prediction.

### Measurement techniques for prostaglandins

Inconsistent study designs and methods greatly limited our ability to compare findings across studies. Up to the late 1990's, researchers most commonly used radioimmunoassay techniques, which can be highly sensitive, but are often limited by the specificity of the antibody used and the potential of antibody cross-reactivity with similar molecules [87]. One study included in

this review developed and reported on a radioimmunoassay for $PGF_{2\alpha}$ with a cross-reactivity with $PGF_{1\alpha}$ of 12.2% [17], which may have obscured patterns in $PGF_{2\alpha}$ and made it difficult to ascertain fine-tuning of the prostaglandin pathway among similar molecules. Furthermore, multiple other studies using radioimmunoassay techniques were unable to differentiate between $PGF_{2\alpha}$ and $PGF_{1\alpha}$, and $PGE_2$ and $PGE_1$ and therefore could only report on levels of PGF and PGE, respectively, making it difficult to compare the results of these studies with others. Lack of specificity and accuracy in these radioimmunoassay techniques may have contributed to the discrepancies across results and highlights the importance of re-visiting dogma in light of novel evidence and technologies. In contrast, the high specificity and sensitivity of mass spectrometry for lipid identification suggests that this method may be more suitable and accurate for measurement of prostaglandins [87]. Additionally, the capability of mass spectrometry to co-assess multiple prostaglandins and metabolites can provide a quantitative profile of prostaglandins before and during labour, as well as identify prostaglandins and/or metabolites not previously measured that may play a role in pregnancy and/or labour [93].

## Considerations among unique biofluids

Among the studies included in this review, the most assayed biofluid was maternal plasma. Although an appealing fluid due to its ability to be sampled relatively easily, results from measurements in plasma were often conflicting, especially among studies that measured primary prostaglandins. Accurate measurement of changes in primary prostaglandin levels in blood is complicated by their rapid metabolism and correspondingly short half-life [94, 95]. This difficulty is further compounded by the production of prostaglandins by platelets that occurs during isolation of plasma and storage of samples [36, 96, 97]. Measurement of plasma PGFM appears to be a good alternative for $PGF_{2\alpha}$, as there is no evidence that this metabolite is formed during sample collection or isolation and therefore may more accurately reflect endogenous prostaglandin production [19]. The primary metabolite of $PGE_2$, however, is chemically unstable [98], which necessitates the measurement of its degradation product, bicyclo-PGEM, for an accurate index of $PGE_2$ production [99]. Therefore, results from early studies measuring primary prostaglandins and/or PGEM in plasma and/or serum should be interpreted with these considerations in mind and future studies in blood should aim to measure PGFM or bicyclo-PGEM as indices of $PGF_{2\alpha}$ or $PGE_2$ production, respectively.

Amniotic fluid lacks prostaglandin metabolizing enzymes [100, 101], which suggests that measurement of the primary prostaglandins in this fluid may be more accurate than in serum or plasma. However, sampling amniotic fluid is more difficult and may introduce infections harmful to the developing fetus, making this fluid impractical as a predictive resource. Additionally, prostaglandin levels vary based on method of collection and region of the amniotic sac [102, 103] which complicates any interpretation of results from studies and limits the clinical utility of an amniotic fluid test for prediction of preterm labour.

Measurement of the main urinary metabolite of $PGF_{2\alpha}$ may be preferable to measuring PGFM in plasma or serum in some cases, as a significant portion of circulating $PGF_{2\alpha}$ is eventually excreted into the urine [104]. In the present investigation, we identified only nine studies that assayed urine, and we suggest that the presence of urinary metabolites of prostaglandins during pregnancy and labour merits further study.

## Demographic and clinical information

Among the articles included in this review, we noted that very few provided complete demographic and clinical information on their participants. Factors including age, race/ethnicity, membrane status, and gravidity/parity may impact prostaglandins levels and a lack of

consideration for these variables may obscure patterns of prostaglandin levels throughout pregnancy and labour. Complete descriptions of gestational age groups and clearly defined outcomes for both term and preterm labour would additionally make studies more easily comparable. As well, preterm labour is generally defined as labour occurring before 37 weeks gestation, however the pathophysiological processes involved in extreme preterm birth (<28 weeks) may vary dramatically from those near term [105]. Therefore, stratification of outcome groups based on gestational age at delivery may be more informative, though would require larger sample sizes to maintain statistical power.

### Role for other prostaglandins

While prostaglandins of the E and F series are most clinically targeted for labour management, there is evidence to suggest that other members of the prostaglandin family may play a role in pregnancy and labour. For example, $PGD_2$ has been shown to increase uterine contractility and blood flow in various mammals [106–108] and is associated with cervical dilation in humans [63]. Two metabolites of $PGD_2$, $9\alpha,11\beta$-$PGF_2$ and $PGJ_2$, were each identified only once among the articles included in this review and were both reported to increase with term labour [84, 87]. These metabolites may be of interest to future researchers, as the development of new methodologies such as mass spectrometry have allowed for more accurate and sensitive measurements of select members of the prostaglandin pathway.

### Limitations

The main limitation of this review is that only studies in English or with an available English translation were included, which may have resulted in missing some relevant articles. However, current resources limited the feasibility of including non-English studies.

### Conclusion

We have identified evidence to suggest that prostaglandin levels, particularly within the PGE and PGF families, do increase in some biofluids during pregnancy and labour. However, changing prostaglandin levels throughout pregnancy and labour are likely highly complex and warrant further investigation, including serial measurements with more precise methodologies in higher-powered studies. Two important limitations identified in this review are the lack of data on the complexity of the prostaglandin pathway outside of the PGE and PGF families and the inherent difficulty in measuring primary prostaglandins in blood, due to their short half-lives in this biofluid. With the advent of i) new methodologies that can assess multiple prostaglandins and metabolites together, ii) a more developed understanding of the range of prostaglandins and iii) a better understanding of the heterogeneous nature of term and preterm labour, future studies that take each of these parameters into account in their study design will help provide further insight into the changing levels of prostaglandins in pregnancy and labour.

### Supporting information

**S1 Table. EMBASE search terms.**
(DOCX)

**S2 Table. MEDLINE search terms.**
(DOCX)

**S3 Table. Quality assessment scores.**
(DOCX)

**S1 Checklist. PRISMA 2020 checklist.**
(DOC)

## Acknowledgments

We would like to acknowledge SL Wood for his help with this systematic review.

## Author Contributions

**Conceptualization:** Eilidh M. Wood, Kylie K. Hornaday, Donna M. Slater.

**Investigation:** Eilidh M. Wood, Kylie K. Hornaday.

**Supervision:** Donna M. Slater.

**Visualization:** Eilidh M. Wood, Kylie K. Hornaday, Donna M. Slater.

**Writing – original draft:** Eilidh M. Wood, Kylie K. Hornaday, Donna M. Slater.

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
