## [Decision Letter · Decision Letter 0]

30 Sep 2021

PONE-D-21-27902Prostaglandins in biofluids in pregnancy and labour: a systematic reviewPLOS ONE

Dear Dr. Slater,

Thank you for submitting your manuscript to PLOS ONE. After careful consideration, we feel that it has substantial merit and the minor revisions requested by the Reviewers will further help to meet PLOS ONE’s publication criteria. Therefore, we invite you to submit a revised version of the manuscript that addresses the points raised during the review process.

We look forward to receiving your revised manuscript.

Kind regards,

Tamas Zakar

Academic Editor

PLOS ONE

Journal Requirements:

2. We note that you have included the phrase “date not shown” in your manuscript. Unfortunately, this does not meet our data sharing requirements. PLOS does not permit references to inaccessible data. We require that authors provide all relevant data within the paper, Supporting Information files, or in an acceptable, public repository. Please add a citation to support this phrase or upload the data that corresponds with these findings to a stable repository (such as Figshare or Dryad) and provide and URLs, DOIs, or accession numbers that may be used to access these data. Or, if the data are not a core part of the research being presented in your study, we ask that you remove the phrase that refers to these data.

3. We note that this manuscript is a systematic review or meta-analysis; our author guidelines therefore require that you use PRISMA guidance to help improve reporting quality of this type of study. Please upload copies of the completed PRISMA checklist as Supporting Information with a file name “PRISMA checklist”

Reviewers' comments:

Reviewer's Responses to Questions

**Comments to the Author**

1. Is the manuscript technically sound, and do the data support the conclusions?

Reviewer #1: Yes

Reviewer #2: Yes

2. Has the statistical analysis been performed appropriately and rigorously? 

Reviewer #1: Yes

Reviewer #2: Yes

3. Have the authors made all data underlying the findings in their manuscript fully available?

Reviewer #1: Yes

Reviewer #2: Yes

4. Is the manuscript presented in an intelligible fashion and written in standard English?

Reviewer #1: Yes

Reviewer #2: Yes

5. Review Comments to the Author

Reviewer #1: ORIGINALITY: This is a systematic review of studies measuring endogenous prostaglandins in blood, amniotic fluid and urine during pregnancy and spontaneous labour. Previous systematic reviews have focussed on exogenous prostaglandins for the induction of labour (eg. Alfirevic et al BMJ 2015;350:h217 and BJOG 2016;123:1462-70). Other recent reviews of the mechanism of parturition have been non-systematic and more wide-ranging (eg Vannuccini et al, Annales d’Endocrinologie 2016;77:105-13, and Mendelson et al, J Steroid Biochem Mol Biol 2017;170:19-27). These show little or no overlap with the present study and I can find no similar systematic reviews. This paper is therefore original.

SCIENTIFIC RELIABILITY: The authors have used the standard methodology of systematic reviews. The inclusion and exclusion criteria are appropriate, as are the quality assessment criteria. I am not aware of any studies that have been missed, and the Discussion seems to me to be balanced and objective.

At line 286 the authors refer to the short half-life of prostaglandins in blood, which I think is the major factor limiting the clinical application of these studies. This issue is discussed clearly in this section of the paper (lines 281-308) but might be worth mentioning again in the Conclusion.

CLINICAL IMPORTANCE: The purpose of this review is to provide a basis for future studies aimed at improving the clinical monitoring of pregnancies. The paper in itself does not have direct clinical relevance but I believe it is very helpful in giving an authoritative overview of what we know (and more importantly, don’t know) about endogenous prostaglandins in pregnancy. It fills an important gap in the literature and I believe it will be widely cited.

OTHER COMMENTS: The paper satisfies the criteria for publication in PLOS ONE. It is original and as far as I know the results have not been published elsewhere. The review is described in sufficient detail and the conclusions are appropriate. The English is excellent and the data availability standards have been met.

Reviewer #2: In this study, the authors systematically searched and summarized the existing scientific literature related to detection of prostaglandins levels in biofluids of pregnancy women, including amniotic fluid, plasma and urine, which in order to determine how biofluid levels of prostaglandins change throughout pregnancy before and during labor. This is a well-written review with comprehensive summary. Although the methods of detecting prostaglandins were diverse in different studies, this manuscript will be improved if the authors can summarize the concentrations of prostaglandins measured under the same methods.

6. PLOS authors have the option to publish the peer review history of their article (what does this mean?). If published, this will include your full peer review and any attached files.

Reviewer #1: No

Reviewer #2: No

---

## [Author Response · Author response to Decision Letter 0]

26 Oct 2021

Comment 1: Please ensure that your manuscript meets PLOS ONE's style requirements, including those for file naming. 

Response: To ensure that our manuscript meets PLOS ONE’s style requirements the beginning line of each paragraph has been indented, author names on the title page have been reformatted (line 7, page 1), and postal codes and street addresses have been removed from the author affiliations on the title page (lines 9-13, page 1). In addition, the file S1_File has been renamed “PRISMA checklist” in accordance with journal requirement.

Comment 2: We note that you have included the phrase “date not shown” in your manuscript. Unfortunately, this does not meet our data sharing requirements. PLOS does not permit references to inaccessible data. We require that authors provide all relevant data within the paper, Supporting Information files, or in an acceptable, public repository. Please add a citation to support this phrase or upload the data that corresponds with these findings to a stable repository (such as Figshare or Dryad) and provide and URLs, DOIs, or accession numbers that may be used to access these data. Or, if the data are not a core part of the research being presented in your study, we ask that you remove the phrase that refers to these data.

Response: We have removed the phrase “not shown” from the manuscript, as it referred to data that is not shown in Figure 2, but which is included in Table 2 and within the manuscript text. 

Comment 3: We note that this manuscript is a systematic review or meta-analysis; our author guidelines therefore require that you use PRISMA guidance to help improve reporting quality of this type of study. Please upload copies of the completed PRISMA checklist as Supporting Information with a file name “PRISMA checklist”

Response: Our completed PRISMA checklist was previously named S1_File and we have renamed it as “PRISMA checklist” and uploaded it as supporting information.

Comment 4: Please review your reference list to ensure that it is complete and correct. If you have cited papers that have been retracted, please include the rationale for doing so in the manuscript text, or remove these references and replace them with relevant current references. Any changes to the reference list should be mentioned in the rebuttal letter that accompanies your revised manuscript. If you need to cite a retracted article, indicate the article’s retracted status in the References list and also include a citation and full reference for the retraction notice.

Response: We have reviewed our reference list and confirmed that we have not referenced any retracted articles. To ensure that the reference list is formatted correctly based on the Vancouver reference style guidelines, we have changed all journal names to their abbreviated versions. Additionally, the reference list was re-ordered slightly to ensure that all references are numbered in the order that they appear in the text (i.e., the reference for Ananth et al. (2013) was moved to position 105 in the reference list (originally position 108) and all the following references moved up one position in the reference list).

Reviewer’s Comments to the Authors

Reviewer 1:

Comment 1: At line 286 the authors refer to the short half-life of prostaglandins in blood, which I think is the major factor limiting the clinical application of these studies. This issue is discussed clearly in this section of the paper (lines 281-308) but might be worth mentioning again in the Conclusion.

Response: We agree with this comment – thank you. Therefore, we have added lines 363-366 (page 29) to the conclusion to reiterate the major limitations identified in this review, including the inability to measure primary prostaglandins in blood due to their short half-lives.

Reviewer 2:

Comment 1: Although the methods of detecting prostaglandins were diverse in different studies, this manuscript will be improved if the authors can summarize the concentrations of prostaglandins measured under the same methods.

Response: We agree with this suggestion – thank you. Therefore, we have added Table 3 on pages 18-20, which includes the range of prostaglandin concentrations reported under each measurement technique. Lines 156-157 on page 9 refer to this table.

---

## [Decision Letter · Decision Letter 1]

3 Nov 2021

Prostaglandins in biofluids in pregnancy and labour: a systematic review

PONE-D-21-27902R1

Dear Dr. Slater,

We’re pleased to inform you that your manuscript has been judged scientifically suitable for publication and will be formally accepted for publication once it meets all outstanding technical requirements.

Kind regards,

Tamas Zakar

Academic Editor

PLOS ONE

Additional Editor Comments (optional):

Reviewers' comments:

Reviewer's Responses to Questions

**Comments to the Author**

1. If the authors have adequately addressed your comments raised in a previous round of review and you feel that this manuscript is now acceptable for publication, you may indicate that here to bypass the “Comments to the Author” section, enter your conflict of interest statement in the “Confidential to Editor” section, and submit your "Accept" recommendation.

Reviewer #1: All comments have been addressed

Reviewer #2: All comments have been addressed

2. Is the manuscript technically sound, and do the data support the conclusions?

Reviewer #1: Yes

Reviewer #2: Yes

3. Has the statistical analysis been performed appropriately and rigorously? 

Reviewer #1: Yes

Reviewer #2: Yes

4. Have the authors made all data underlying the findings in their manuscript fully available?

Reviewer #1: Yes

Reviewer #2: Yes

5. Is the manuscript presented in an intelligible fashion and written in standard English?

Reviewer #1: Yes

Reviewer #2: Yes

6. Review Comments to the Author

Reviewer #1: (No Response)

Reviewer #2: The authors have adequately addressed all comments.

7. PLOS authors have the option to publish the peer review history of their article (what does this mean?). If published, this will include your full peer review and any attached files.

Reviewer #1: No

Reviewer #2: No

---

## [Editor Report · Acceptance letter]

9 Nov 2021

PONE-D-21-27902R1 

Prostaglandins in biofluids in pregnancy and labour: a systematic review 

Dear Dr. Slater:

I'm pleased to inform you that your manuscript has been deemed suitable for publication in PLOS ONE. Congratulations! Your manuscript is now with our production department. 

Kind regards, 

on behalf of

Dr Tamas Zakar 

Academic Editor

PLOS ONE